# Robust Reward Sequence Modeling with Multi-Scale Consistency for Model-Based Reinforcement Learning

## Abstract

We propose a novel framework for reliable reward modeling in model-based reinforcement learning, built on top of Mamba-based sequence models. While prior work suffers from cumulative error over long rollouts due to decoding only immediate rewards from the latent dynamics, our approach trains an ensemble of multi-horizon reward heads that each predict the cumulative return over different horizons. To tie these predictions together, we introduce a cross-horizon consistency regularization to encourage the difference between any two heads to match the prediction of their gap head. We further add a chunk-level reward model that summarizes rewards over non-overlapping blocks, and enforce consistency between chunk and per-step predictions for smoother estimates. During imagination, we dynamically select the reward heads with the lowest predictive uncertainty to guide policy rollouts, and combine these multi-scale predictions with the standard $\lambda$-return during value estimation. This design ensures that more accurate, well-conditioned reward estimates directly shape policy learning. We integrate our method into Drama, a state-of-the-art Mamba-enabled model-based agent, and evaluate on the *Atari 100k* benchmark. Compared to the single-head baseline, our multi-scale, cross-horizon consistency approach reduces reward prediction error by $47\%$ on average and yields higher or comparable game scores across the suite. Our results demonstrate that explicitly modeling and regularizing rewards at multiple temporal scales and carefully enlisting the most confident predictions improve both the fidelity of imagined rollouts and the policy performance.

## 1 Introduction

Model-based reinforcement learning (MBRL) Moerland et al. (2023); Micheli et al. (2023a); Hafner et al. (2023); Kaiser et al. (2020) is known for its virtues in terms of policy robustness, sample efficiency, and interpretability compared to model-free RL. This learning paradigm explicitly learns a predictive world model to estimate the dynamics and rewards of the underlying environment, which offers a rational and highly interpretable policy derived from the learned decision model. Moreover, the estimated model enables agents to "imagine" artificial trajectories internally for value estimation and policy optimization without having to collect new real-world data. This capability dramatically improves sample efficiency. Yet classical MBRL approaches rely on short-horizon, one-step transition models or Gaussian process approximations, making them struggle to capture long-term dependencies and suffer from compounding errors during rollouts.

To tackle this challenge, sequence modeling techniques Chen et al. (2021); Wang et al. (2023); Zhuang et al. (2024) emerge. They treat planning trajectories as token sequences Fang et al. (2019), enabling the model to capture temporal dependencies and predict future states, rewards, and other relevant signals from learned latent representations of the agent's history. Transformer-based dynamics models Micheli et al. (2023b); Hafner et al. (2023) can flexibly handle hundreds of past timesteps, thereby capturing long-range correlations in both state and reward signals. Unfortunately, their quadratic time and memory complexity in sequence length make them intractable for long rollouts or high-resolution inputs. In contrast, the Mamba family Gu & Dao (2024); Dao & Gu (2024) of state-space models (SSMs) achieves linear scaling in sequence length while retaining the ability to summarize distant context via continuous-time convolutions. By embedding SSMs within modern

variational or hybrid frameworks Ota (2024); Lv et al. (2024); Huang et al. (2024), one can train compact, expressive world models Wang et al. (2025) that support sample-efficient planning on high-dimensional tasks such as *Atari 100k*.

Despite advances in Mamba-based MBRL, they suffer from unreliable reward modeling over extended horizons. For example, we evaluate Drama Wang et al. (2025), a state-of-the-art (SOTA) Mamba-based MBRL method in an Atari game Alien, and illustrate its average cumulative reward prediction error over time in Fig. 1. We observe that the error spikes after the 14th step, particularly with learned policy where errors compound more severely. This challenge stems from three key limitations. First, conven-

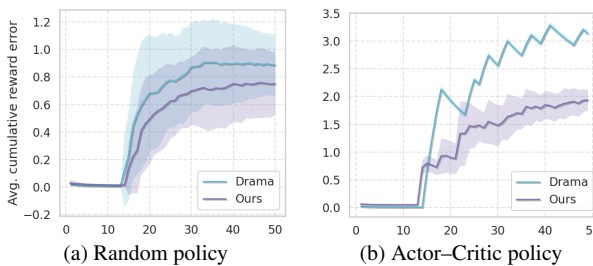

(a) Random policy      (b) Actor–Critic policy

Figure 1: Average cumulative reward prediction errors by random action selection (left) and policy learned along with the world model (right) in Alien. In both cases, the proposed approach effectively constrains the error over extended horizons, showing an evident performance gain over the baseline.

tional reward decoders only predict immediate rewards, causing small errors to accumulate without any temporal consistency constraints across different prediction horizons. Second, even with multi-step reward encoders, the lack of cross-horizon consistency leads to divergent estimates that corrupt downstream policy learning. Third, focusing exclusively on fine-grained per-timestep predictions overlooks coarse-grained reward patterns that could provide more stable long-term guidance. We argue that these complementary reward modeling deficiencies ultimately compromise policy quality.

In this paper, we propose Robust Reward Modeling with Multi-Scale Consistency, a novel enhancement to Mamba-based MBRL that tackles reward-prediction drift at its roots. Here are our core ideas. **(1) Multi-horizon reward ensemble.** We train an ensemble of reward heads, each predicting the sum of future rewards over a different horizon $h \in \mathcal{H} = \{h_1, h_2, \ldots, h_K\}$. **(2) Cross-Horizon consistency.** We introduce a regularizer that enforces $\hat{r}_t^{(h_i + h_j)} \approx \hat{r}_t^{(h_i)} + \hat{r}_{t+h_1}^{(h_j)}$, where $\hat{r}_t^{(h)}$ denotes the predicted cumulative reward over the next $h$ steps at timestep $t$. **(3) Chunk-level reward head and slow-fast consistency.** We augment the ensemble with a low-frequency, chunk-stride head that predicts entire $C$-step sums from a single latent state. This head provides robust coarse rewards by explicitly modeling reward patterns at a broader temporal scale, enabling the agent to capture reward dynamics that might be missed by fine-grained predictions. Finally, we blend the standard generalized advantage estimator (GAE) Schulman et al. (2016) $\lambda$-return with the introduced enhancements, ensuring that consistent multi-scale rewards directly guide policy and value learning. As shown in Fig. 1, our approach demonstrates a noticeably better-bounded cumulative error.

We implement our method atop the Drama Wang et al. (2025) framework and evaluate on the *Atari 100k* benchmark. Empirically, we observe a $47\%$ reduction in mean absolute reward-prediction error across horizons, and up to $25\%$ improvement in median normalized score over Drama. These gains demonstrate that enforcing multi-scale and cross-horizon consistency in reward modeling not only tames error accumulation but also translates into more robust policy learning under limited data.

## 2 METHODOLOGY

In this section, we first demonstrate the pipeline used to train the latent world model and behavior policy jointly. Then, we walk through the proposed techniques: ensemble multi-step reward estimation, cross-horizon consistency, chunk-level reward estimation, and slow-fast consistency.

### 2.1 GENERAL PIPELINE

We formulate the problem as a Partially Observable Markov Decision Process (POMDP) Cassandra (1998); Kaelbling et al. (1998); Sondik (1971). Unlike its fully observable counterpart, i.e., MDP, a POMDP is technically non-Markovian in the absence of direct access to the underlying physical state. To keep track of the state estimation, one has to maintain the entire history of partial observation and

---

**Algorithm 1** Training procedures for the world model

---

**Require:** Obsv. trajectory $\boldsymbol{\tau}_o$, action trajectory $\boldsymbol{\tau}_a$, reward trajectory $\boldsymbol{\tau}_r$, termination signals $\boldsymbol{\tau}_d$

**Require:** Horizons $\mathcal{H} = \{h_1, \ldots, h_K\}$, chunk size $C$, loss weights $\{\alpha_r^{(h)}\}, \alpha_r^{\text{chunk}}, \beta_{\text{rc}}, \beta_{\text{sfc}}$

1: $\mathcal{L}_{\text{others}}, \mathcal{L}_r^{(1)}, \boldsymbol{\tau}_z \leftarrow \text{DramaBackbone}(\boldsymbol{\tau}_o, \boldsymbol{\tau}_a, \boldsymbol{\tau}_d)$     $\triangleright \mathcal{L}_{\text{others}}$ includes $\mathcal{L}_{\text{recon}}, \mathcal{L}_{\text{dyn}}$, and $\mathcal{L}_{\text{term}}$

2: **for all** $h \in \mathcal{H}$ **do**

3:     $\boldsymbol{\tau}_{r^{(h)}} \leftarrow [\,]$

4:     **for** $t \leftarrow 0 \ldots T$ **do**

5:        $\boldsymbol{\tau}_{r^{(h)}}[t] \leftarrow \sum_{i=0}^{h-1} \gamma^i \boldsymbol{\tau}_r[t+i]$     $\triangleright$ Calculate ground truth cumulative rewards

6:     **end for**

7:     $\boldsymbol{\tau}_{\hat{r}^{(h)}} \leftarrow f_{\hat{R}^{(h)}}(\boldsymbol{\tau}_z), \mathcal{L}_r^{(h)} \leftarrow \mathcal{L}_{\text{SL2H}}(\boldsymbol{\tau}_{\hat{r}^{(h)}}, \boldsymbol{\tau}_{r^{(h)}})$

8: **end for**

9: $\mathcal{L}_{\text{cons}}^{\text{reward}} \leftarrow 0$

10: **for all** $(h_i, h_j) \in \mathcal{H} \times \mathcal{H}$ where $h_i < h_j$ and $h_j - h_i \in \mathcal{H}$ **do**

11:     $\mathcal{L}_{\text{cons}}^{\text{reward}} \leftarrow \mathcal{L}_{\text{cons}}^{\text{reward}} + \frac{1}{T-h_j} \sum_{t=0}^{T-h_j-1} \|\boldsymbol{\tau}_{\hat{r}^{(h_j)}}[t] - (\boldsymbol{\tau}_{\hat{r}^{(h_i)}}[t] + \gamma^{h_i} \boldsymbol{\tau}_{\hat{r}^{(h_j-h_i)}}[t+h_i])\|^2$

12: **end for**

13: $\boldsymbol{\tau}_{r^{\text{chunk}}} \leftarrow [\,], \boldsymbol{\tau}_{\hat{r}^{(C)}} \leftarrow [\,]$

14: **for** $j \leftarrow 0 \ldots \lfloor T/C \rfloor - 1$ **do**

15:     $\boldsymbol{\tau}_{r^{\text{chunk}}}[j] \leftarrow \sum_{i=0}^{C-1} \boldsymbol{\tau}_r[j \cdot C + i]$     $\triangleright$ Sum rewards in non-overlapping chunks

16: **end for**

17: $\boldsymbol{\tau}_{\hat{r}^{\text{chunk}}} \leftarrow f_{\hat{R}^{\text{chunk}}}(\boldsymbol{\tau}_z[:: C])$     $\triangleright$ Predict at chunk boundaries

18: $\mathcal{L}_r^{\text{chunk}} \leftarrow \mathcal{L}_{\text{SL2H}}(\boldsymbol{\tau}_{\hat{r}^{\text{chunk}}}, \boldsymbol{\tau}_{r^{\text{chunk}}})$

19: **for** $j \leftarrow 0 \ldots \lfloor T/C \rfloor - 1$ **do**

20:     $\boldsymbol{\tau}_{\hat{r}^{(C)}}[j] \leftarrow \sum_{i=0}^{C-1} \boldsymbol{\tau}_{\hat{r}^{(1)}}[j \cdot C + i]$     $\triangleright$ Sum consecutive single-step predictions

21: **end for**

22: $\mathcal{L}_{\text{cons}}^{\text{slowfast}} \leftarrow \|\boldsymbol{\tau}_{\hat{r}^{\text{chunk}}} - \boldsymbol{\tau}_{\hat{r}^{(C)}}\|^2$

23: $\mathcal{L}_{\text{total}} \leftarrow \mathcal{L}_{\text{others}} + \sum_{h \in \mathcal{H}} \alpha_r^{(h)} \mathcal{L}_r^{(h)} + \alpha_r^{\text{chunk}} \mathcal{L}_r^{\text{chunk}} + \beta_{\text{rc}} \cdot \mathcal{L}_{\text{cons}}^{\text{reward}} + \beta_{\text{sfc}} \cdot \mathcal{L}_{\text{cons}}^{\text{slowfast}}$

24: $\theta \leftarrow \theta - \eta_\theta \nabla_\theta \mathcal{L}_{\text{total}}$

---

action pairs $\boldsymbol{h}_t = (o_1, a_1, o_2, a_2, \ldots, o_t)$. This particularly suits sequence modeling, especially for Mamba that can capture dependencies in extremely long history with linear complexity.

Our goal is to use a sequence model like Mamba as backbone to explicitly model the underlying dynamics and reward of the target POMDP. We learn a latent state representation $z_t$ that encodes the observation history $\boldsymbol{h}_t$, and model the transition dynamics $p(z_{t+1}|z_t, a_t)$ as well as the reward function $f_R(r_t|z_t, a_t)$. Then, using the artificial data produced by this learned world model, we optimize a policy $\pi(a_t \mid z_t)$ that maximizes expected returns.

We illustrate the proposed framework including world model estimation and behavior policy learning in Fig. 2 and provide the procedures of world model training in Algorithm 1. Our architecture builds upon the Drama framework, which uses a Mamba backbone to govern sequential latent state transitions. The encoder transforms observations into latent representations, while a discrete variational autoencoder (VAE) framework with categorical distributions enables efficient representation learning. For the reward modeling aspects, we enhance the standard single-step reward predictor $f_{\hat{R}}^{(1)}$ with two key components. The first is an ensemble of multi-horizon reward heads $\{f_{\hat{R}}^{(h)}\}$, each predicting cumulative returns over different horizons $h \in \mathcal{H}$, which we constrain with cross-horizon consistency regularization. The second is a chunk-level "slow" reward model $f_{\hat{R}}^{\text{chunk}}$ that operates on non-overlapping blocks of $C$ timesteps, providing a coarse-grained perspective that complements the fine-grained rewards through slow-fast consistency. We demonstrate these components in detail in Sec. 2.2 and Sec. 2.3, respectively.

## 2.2 LONG-HORIZON REWARD REGULARIZATION

In world model estimation, to effectively address the challenge of accumulated error in reward modeling over extended planning horizons, we propose a comprehensive framework with two key components. First, we train an ensemble of specialized reward decoders that estimate cumulative returns over different horizons, creating a more robust predictive mechanism compared to conventional

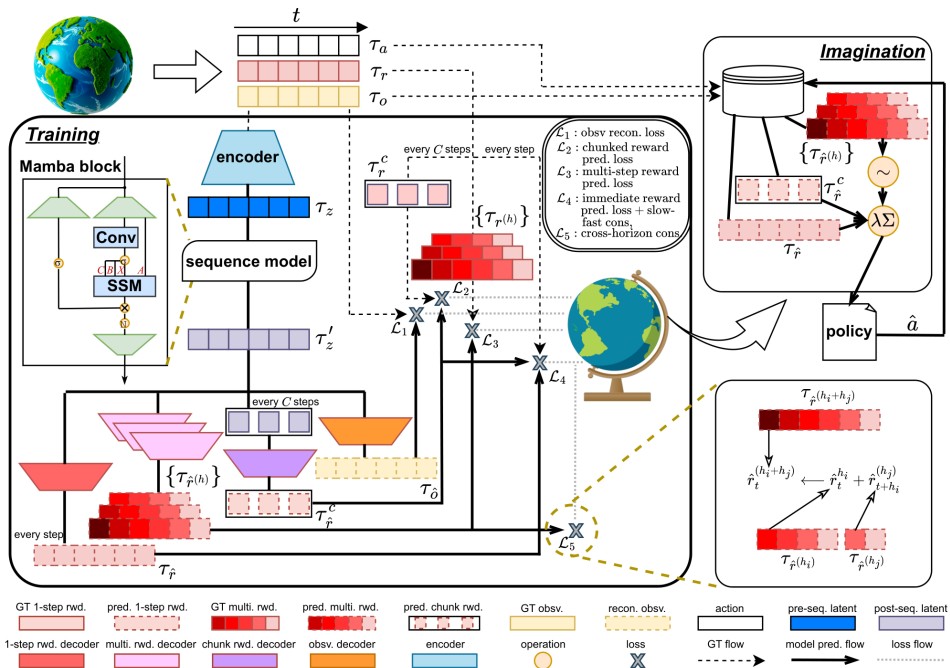

Figure 2: **Illustration of the proposed MBRL framework.** The framework consists of two main modules, the latent world model and behavior policy module. During the joint training, the two modules are connected by imagination, a process in which the agent rolls out using "imagined" dynamics produced by the learned world model rather than real data. This diagram focuses on demonstrating the latent world model estimation and the imagination. In the imagination block, $\sim$ denotes the ensemble decision process for the set of estimated multi-horizon rewards $\{\tau_{\hat{r}^{(h)}}\}$ based on variance. $\lambda\Sigma$ designates weighted sum of multi-scale rewards. The policy icon abstracts an entire policy learning network, where we can use any standard on-policy RL to derive a desired policy.

single-step reward models. Second, we introduce a cross-horizon consistency regularization that enforces agreement between predictions at different temporal scales, ensuring coherence in the model's understanding of reward dynamics. This dual approach improves prediction accuracy across various horizons and provides a principled way to handle uncertainty during imagination, enabling more reliable policy optimization.

**Multi-step reward supervision and uncertainty-aware imagination.** We formulate reward prediction as a multi-scale modeling problem by introducing a family of decoders $\{f_{\hat{R}}^{(h)}\}_{h\in\mathcal{H}}$, each predicting the cumulative return over horizon $h$. For each horizon $h \in \mathcal{H}$, the ground-truth cumulative return at time $t$ is defined as $r_t^{(h)} = \sum_{i=0}^{h-1} \gamma^i r_{t+i}$, representing the discounted sum of rewards over the next $h$ steps. Each decoder head produces an estimate $\hat{r}_t^{(h)}$ from the same latent state output by our sequential backbone. We train these heads using a symmetric log two-hot encoded loss

$$\mathcal{L}_{\text{multi-}r} = \sum_{h\in\mathcal{H}} \alpha_r^{(h)} \mathcal{L}_{\text{SL2H}}\left(\boldsymbol{\tau}_{\hat{r}^{(h)}}, \boldsymbol{\tau}_{r^{(h)}}\right). \tag{1}$$

During the imagination phase, we use an ensemble of reward predictors, each specialized on different horizons. Specifically, for each candidate action sequence, we roll out all of our predictors within the ensemble. Then, according to the variance for the predictions, we select those with lower variance out of others and score a sequence by a weighted sum of their cumulative return predictions via Eq. (2).

$$\boldsymbol{\tau}_{\hat{r}^{\text{ensem}}}[t] = \frac{\sum_{h\in\mathcal{S}_t} w_t^{(h)} \boldsymbol{\tau}_{\hat{r}^{(h)}}[t]}{\sum_{h\in\mathcal{S}_t} w_t^{(h)}}, \quad \text{where} \quad w_t^{(h)} = \frac{1}{\left(\sigma_t^{(h)2} + \epsilon\right)^\beta}. \tag{2}$$

Here, $\sigma_t^{(h)2}$ is the estimated variance of horizon $h$ at timestep $t$. $\epsilon > 0$ is a small constant. $T$ denotes the temperature-scaling parameter that controls how sharply we focus on the most confident heads.

Such a set of jointly trained multi-step reward decoders with uncertainty-aware reward estimation makes planning more robust to noisy reward predictions. If a particular head produces wildly anomalous estimates, it will most likely be filtered out by our variance-based selection. Even for included heads with moderately elevated uncertainty, their influence is automatically down-weighted in the ensemble average. Technically, this dynamic weighting mechanism provides a smooth degradation path that creates a more nuanced and resilient reward estimation system. This ensures that more reliable predictors compensate for less confident ones.

**Cross-horizon-consistent reward estimation.** Adding multiple reward heads inevitably divides the network's capacity across one-step, multi-step, and other objectives, potentially diluting its ability to capture sharp reward spikes that a specialized single-step decoder would detect. To address this trade-off, we introduce cross-horizon consistency regularization that explicitly couples predictions across different time scales. This approach ensures that a reward spike at timestep $t$ is consistently represented in both the one-step head and all longer-horizon heads. By aligning gradient signals across temporal scales, we reduce conflicting updates that might otherwise obscure transient high-magnitude rewards. The result is a model that maintains sensitivity to important reward spikes while simultaneously providing smooth, accurate long-term predictions.

We formulate the cross-horizon consistency and demonstrate its usage in the reward gradient calibration as follows. For some horizon $h$ and $k$, where $h < k$, we define $\hat{r}_{1 \to h} + \hat{r}_{(h+1) \to (k)} \approx \hat{r}_{1 \to k}$ as the consistency between horizon $h$, $h - k$, and $k$. In practice, at any timestep $t$, for every pair of horizons $(h_i, h_j) \in \mathcal{H}$ satisfying the constraint that $h_{j-i} \in \mathcal{H}$ (suppose $i < j$), we enforce the cross-horizon reward prediction consistency by minimizing the following loss:

$$\mathcal{L}_{\text{cons}}^{\text{reward}} = \sum_{\{h_i, h_j, h_{j-i}\} \subseteq \mathcal{H}} \frac{1}{T - h_j} \sum_{t=0}^{T - h_j - 1} \|\boldsymbol{\tau}_{\hat{r}^{(h_j)}}[t] - (\boldsymbol{\tau}_{\hat{r}^{(h_i)}}[t] + \gamma^{h_i} \boldsymbol{\tau}_{\hat{r}^{(h_j - h_i)}}[t + h_i])\|^2. \quad (3)$$

**Horizon set selection.** Different environments may have different underlying reward feedback patterns. A uniformly selected set of horizons for multi-horizon reward decoders cannot work equally well in all environments. Therefore, we need to choose a dedicated set for each environment, or those sharing similar reward patterns. Using games in the *Atari 100k* benchmark as examples. Games like *Boxing* and *Pong*, which exhibit a denser reward feedback in relatively rapid cycles, typically prefer denser and shorter horizons. In contrast, games like *Freeway* and *PrivateEye*, which have more complex mechanisms and demonstrate slower and sparser reward feedback, are usually in favor of sparser and longer horizons. A reasonable horizon selection aware of reward patterns is another key to accurately model the reward function and thereby boost the policy quality. We discuss in detail the impact of the selection of the horizon set on policy performance in Sec. A.2.

## 2.3 SLOW-FAST REWARD AND VALUE ESTIMATION

To reduce the high-frequency noise and improve long-horizon coherence in our reward estimates, we augment the current "fast" world model estimator with a "slow", chunk-level predictor. It predicts the cumulative return over non-overlapping blocks of $C$ timesteps. Specifically, given the latent feature sequence $\boldsymbol{\tau}_z = \{\boldsymbol{z}_t\}_{t=1}^{T}$, the chunk-level predictor produces:

$$\hat{r}_t^{\text{chunk}} = f_{\text{reward}}^{\text{chunk}}(\boldsymbol{z}_{t+C-1}) \approx \sum_{k=0}^{C-1} r_{t+k}, \quad (4)$$

where $r_{t+k}$ is the true reward at timestep $t + k$. We train the chunk-level predictor with the same sym-log-two-hot loss $\mathcal{L}_{\text{SL2H}}$,

$$\mathcal{L}_r^{\text{chunk}} = \mathcal{L}_{\text{SL2H}}(\boldsymbol{\tau}_{\hat{r}^{\text{chunk}}}, \boldsymbol{\tau}_{r^{\text{chunk}}}). \quad (5)$$

To enforce consistency between scales, we add a slow-fast consistency loss:

$$\mathcal{L}_{\text{cons}}^{\text{slowfast}} = \frac{1}{\lfloor T/C \rfloor} \sum_{j=0}^{\lfloor T/C \rfloor - 1} \|\boldsymbol{\tau}_{\hat{r}^{\text{chunk}}}[j] - \boldsymbol{\tau}_{\hat{r}^{(C)}}[j]\|^2, \quad (6)$$

which penalizes discrepancies between the chunked prediction and the $C$-step sum of per-step returns. During imagination, the slow head provides coarse reward signals that complement fine-grained fast

estimates, providing a higher-level view for long-horizon planning, similar in sense to the hierarchical, multi-grained mechanisms.

Despite both capturing multi-step cumulative returns and imposing consistency across the steps, the chunk-level module has major differences than the multi-horizon reward predictors. A $k$-horizon head from the ensemble makes overlapping, per-step forecasts $\hat{r}_t^{(k)}$ for each $t$. It reuses the same latent feature $z_t$ that the single-step head $f_{\hat{R}}^{(1)}$ uses, but trained to predict $\sum_{i=0}^{C-1} r_{t+i}$. In contrast, the chunk-level head partitions the trajectory into non-overlapping windows of length $C$. It pools the entire chunk's information into a single feature vector, $z_{\text{chunk}} = g\left(\{z_{t:i\in[t,t+C-1]}\}\right)$, and then predicts the return of that chunk at $t + C - 1$. This encourages the model to learn a coherent, low-frequency summary representation over the whole block, instead of trying to "stretch" a single-step latent feature to make multi-step forecasts at every time point. Empirically, this slow-fast interplay reduces cumulative reward prediction error and mitigates drift in long-horizon rollouts, yielding more robust latent dynamics and improved downstream policy learning.

## 3 EXPERIMENTS

### 3.1 BENCHMARK AND BASELINES

We evaluate our method on the *Atari 100k* benchmark Kaiser et al. (2020), a widely adopted suite that measures sample efficiency by limiting each agent to just $100,000$ environment interactions across 26 Atari games. Performance is reported in terms of human-normalized scores and median normalized score, emphasizing rapid learning under severe data constraints.

We compare against a range of model-based and sequence-modeling approaches. SimPle Kaiser et al. (2020) is the first to demonstrate world model planning on Atari under low budgets. SPR Schwarzer et al. (2021) enhances world models with self-predictive auxiliary losses. IRIS Micheli et al. (2023a) combines contrastive representation learning with reconstruction to improve sample efficiency. STORM is a stochastic Transformer-based world model optimized for high-variance environments. DreamberV3 Hafner et al. (2023) is the latest iteration of the Dreamer family with classifier-free dynamics modeling. HarmonyDream Ma et al. (2024) harmonizes multi-task behaviors inside a shared world model. Drama Wang et al. (2025) is our baseline built on the Mamba-enabled Drama framework. All experiments are conducted under the same interaction step constraint, using identical preprocessing, action repeats, and evaluation protocols, allowing a fair comparison.

### 3.2 RESULTS AND DISCUSSION

**Reward modeling.** We use the average cumulative reward prediction error per horizon (Eq. (7)) as the metric to evaluate the quality of the reward modeling. It captures how accurately each model predicts the rewards over time, with lower values indicating better performance.

$$\text{avg\_err}\,(N) = \frac{1}{|\mathcal{E}_N|} \sum_{e \in \mathcal{E}_N} \left(\frac{1}{N} \sum_{t=1}^{N} |\hat{r}_{e,t} - r_{e,t}|\right). \qquad (7)$$

As shown in Fig. 3 top row, our approach consistently achieves significantly smaller errors than Drama across the selected Atari game environments. First, we observe a $47\%$ reduction in mean absolute reward prediction error on all horizons compared to the baseline Drama, demonstrating the effectiveness of our multi-scale consistency approach. Moreover, the error curves show that our method not only achieves lower error but also converges earlier during training, suggesting that the multi-horizon ensemble and consistency constraints help the model learn more accurate reward predictions with fewer environment interactions. The results demonstrate that enforcing multi-scale and cross-horizon consistency in reward modeling effectively tames error accumulation over extended horizons. This is particularly evident in the flatter error curves of our method compared to Drama's steeper error increases. While the improvement pattern is consistent across environments, the magnitude varies.

The plots clearly show that as the prediction horizon increases, the error gap between our method and Drama widens, highlighting how our approach specifically addresses the challenge of long-horizon reward modeling. The shaded regions represent confidence intervals, which demonstrates not only

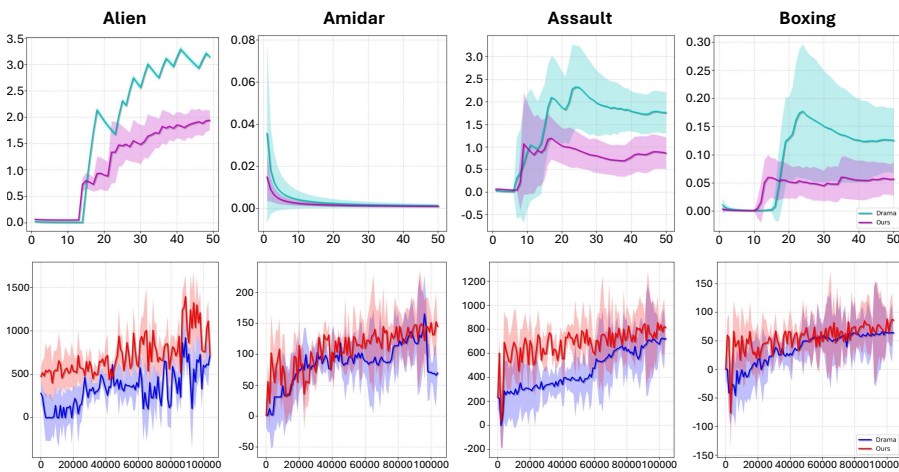

Figure 3: **Visualization of experimental results regarding performance comparison between Drama and our approach.** *Top row*: Average cumulative reward prediction errors against horizons in four Atari environments. It is evident that our approach results in significantly smaller prediction errors than those of the baseline Drama in all environments and converges earlier. *Bottom row*: Online evaluation in gained scores over training steps. Across all four games, our approach learns faster and attains higher overall scores than Drama. This alignment between reduced long-horizon reward error and accelerated policy improvement underscores how our multi-horizon consistency regularizer and chunk-level reward head stabilize imagined rollouts, yielding faster, more reliable learning.

lower mean error but also reduced variance of our approach than Drama in predictions. This indicates more stable and reliable reward modeling. These improvements in reward prediction directly translate to better policy learning, as shown in the bottom row of Fig. 3, where our approach consistently learns faster and achieves higher scores across multiple environments.

**Behavior policy optimization.** The effectiveness of our approach extends beyond improved reward estimation to enhanced policy learning. As illustrated in the bottom row of Fig. 3, our approach consistently outperforms Drama across all environments in terms of learning speed and final performance. Firstly, our method exhibits faster learning progress during the early stages of training. This acceleration is attributed to accurate and consistent reward predictions provided by our multi-scale reward modeling framework. They enable more effective policy optimization from the beginning of training. Secondly, across the four environments, our approach achieves higher final scores by the end of training. This implies that the benefits of improved reward modeling persist throughout the learning process. Finally, the games with the largest improvements in reward prediction also tend to show the most significant gains in policy performance, highlighting the relevance between reward modeling quality and policy effectiveness.

Tab. 1 provides a thorough evaluation of our approach against multiple SOTA baselines on the *Atari 100k* benchmark. The result reveals several important findings. The most evident one is that our approach achieves impressive results across the 26 Atari games in the benchmark, with particularly strong performances in games like Asterix (1681), BattleZone (13250), CrazyClimber (90212), and Hero (10196). Compared with Drama, our method shows significant improvements in most games, with an average improvement of $10.48\%$ in mean and $100\%$ in median normalized score. This substantial improvement validates the effectiveness of our approach in enhancing MBRL.

Our approach achieves competitive or superior results compared to other leading methods such as DreamerV3, HarmonyDream, and STORM across many games. In several challenging environments like Asterix, CrazyClimber, and RoadRunner, our method establishes new state-of-the-art results. The normalized mean and median scores at the bottom of Table 1 provide a standardized comparison across methods. Our approach achieves strong performance on both metrics, demonstrating its general effectiveness across the diverse set of games in the benchmark. The results reveal that our method particularly excels in games requiring long-horizon planning and temporal consistency, such as Hero and CrazyClimber, where reward prediction accuracy over extended sequences is crucial.

Table 1: Comparison for our approach and the baselines on *Atari 100k* benchmark. We highlight the highest, second highest, and third highest scores for each game. Each number is an average over 5 independent seeds.

| Game | Random | Human | SimPLe | SPR | IRIS | STORM | DreamerV3 | HarmonyDream | DramaXS | Ours |
|---|---|---|---|---|---|---|---|---|---|---|
| Alien | 228 | 7128 | 617 | 842 | 420 | **984** | 959 | 890 | 820 | 912 |
| Amidar | 6 | 1720 | 74 | 180 | 143 | **205** | 139 | 141 | 131 | 150 |
| Assault | 222 | 742 | 527 | 566 | **1524** | 801 | 706 | 1003 | 539 | 805 |
| Asterix | 210 | 8503 | 1128 | 962 | 854 | 1028 | 932 | 1140 | 1632 | **1681** |
| BankHeist | 14 | 753 | 34 | 345 | 53 | 641 | 649 | **1069** | 137 | 416 |
| BattleZone | 2360 | 37188 | 4031 | 14834 | 13074 | 13540 | 12250 | **16456** | 10860 | 13250 |
| Boxing | 0 | 12 | 8 | 36 | 70 | **80** | 78 | **80** | 78 | **80** |
| Breakout | 2 | 30 | 16 | 20 | **84** | 16 | 31 | 53 | 7 | 17 |
| ChopperCmd | 811 | 7388 | 979 | 946 | 1565 | **1888** | 420 | 1510 | 1642 | 1651 |
| CrazyClimber | 10780 | 35829 | 62584 | 36700 | 59324 | 66776 | **97190** | 82739 | 83931 | 90212 |
| DemonAttack | 152 | 1971 | 208 | 518 | **2034** | 165 | 303 | 203 | 201 | 361 |
| Freeway | 0 | 30 | 17 | 19 | 31 | **34** | 0 | 0 | 15 | 23 |
| Frostbite | 65 | 4335 | 237 | 1171 | 259 | **1316** | 909 | 679 | 785 | 838 |
| Gopher | 258 | 2412 | 597 | 661 | 2236 | 8240 | 3730 | **13043** | 2757 | 3342 |
| Hero | 1027 | 30826 | 2657 | 5859 | 7037 | 11044 | 11161 | **13378** | 7946 | 10196 |
| Jamesbond | 29 | 303 | 100 | 366 | 463 | **509** | 445 | 317 | 372 | 485 |
| Kangaroo | 52 | 3035 | 51 | 3617 | 838 | 4208 | 4098 | **5118** | 1384 | 2158 |
| Krull | 1598 | 2666 | 2205 | 3682 | 6616 | 8413 | 7782 | 7754 | **9693** | 9374 |
| KungFuMaster | 258 | 22736 | 14862 | 14783 | 21760 | **26183** | 21420 | 22274 | 23920 | 23906 |
| MsPacman | 307 | 6952 | 1480 | 1318 | 999 | **2673** | 1372 | 1681 | 2270 | 2374 |
| Pong | -21 | 15 | 13 | -5 | 15 | 11 | 18 | **19** | 15 | 18 |
| PrivateEye | 25 | 69571 | 35 | 86 | 100 | **7781** | 882 | 2932 | 90 | 226 |
| Qbert | 164 | 13455 | 1289 | 866 | 746 | **4522** | 3405 | 3933 | 796 | 927 |
| RoadRunner | 12 | 7845 | 5641 | 12213 | 9615 | **17564** | 15565 | 14646 | 14020 | 14315 |
| Seaquest | 68 | 42055 | **683** | 558 | 661 | 525 | 618 | 665 | 497 | 572 |
| UpNDown | 533 | 11693 | 3350 | 10859 | 3546 | 7985 | 7667 | **10874** | 7387 | 7892 |
| Normalized Mean (%) | 0 | 100 | 33 | 62 | 105 | 127 | 125 | **136** | 105 | 116 |
| Normalized Median (%) | 0 | 100 | 13 | 40 | 29 | 58 | 49 | **67** | 27 | 54 |

## 3.3 ABLATION STUDY

To analyze the individual contributions of each component in our approach, we conduct a comprehensive ablation study. Fig. 4 presents the results for the game Alien, with the subfigure on the left showing the impact on cumulative reward prediction error and that on the right the effects on policy performance. We systematically evaluated the three core components of our framework.

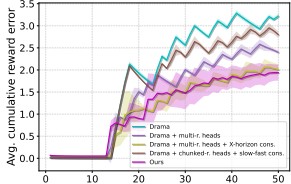

(a) Cumulative returns.

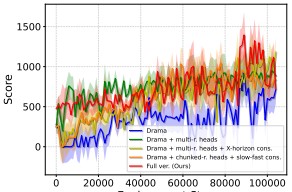

(b) Online policy evaluation.

Figure 4: Ablation study results demonstrating the incremental contribution of each component in reducing reward prediction error and improving policy performance.

**Multi-step reward decoder ensemble.** Adding the multi-step reward heads to Drama provides significant improvement. As shown in Fig. 4a, this modification alone reduces the average cumulative reward prediction error by approximately 25% across horizons. The error reduction is particularly noticeable as the prediction horizon increases beyond 20 steps, demonstrating the ensemble's effectiveness at mitigating error accumulation over longer time scales. On the policy learning side (Fig. 4b), the multi-step reward ensemble leads to improved learning dynamics with faster initial progress and higher intermediate scores. This confirms that having dedicated predictors for different horizons helps the agent better anticipate the consequences of its actions over varying time scales, leading to more informed decision-making. However, the multi-step heads alone, while beneficial, still exhibit increasing divergence at longer horizons, indicating that simply adding multiple prediction targets without enforcing consistency between them has limitations.

**Cross-horizon consistency.** Building upon the multi-step reward heads, adding the cross-horizon consistency regularizer further reduces prediction errors across all horizons. The consistency enforces agreement between different horizon-length predictions, resulting in a more coherent internal model of reward dynamics. Fig. 4a shows that the cross-horizon consistency reduces the prediction error by an additional 15% compared to using just the multi-step heads, with the improvement becoming more pronounced at longer horizons (beyond 30 steps). This indicates that the consistency regularization is particularly effective at constraining drift in long-horizon predictions. The policy performance

in Fig. 4b reflects this improvement, with more stable learning progress and higher scores in the middle stages of training. The cross-horizon consistency helps the agent develop a more coherent understanding of long-term consequences, enabling it to make more consistent policy decisions.

**Chunk-level coarse-grained reward modeling and slow-fast consistency.** This pair of components contribute less to the reduction of cumulative reward error and policy performance gain. As shown in Fig. 4a, adding this pair only to Drama leads to the second highest error, while the earned score is the second lowest. The chunk-level modeling and slow-fast consistency still provide a complementary signal that stabilizes long-horizon prediction, but the effectiveness is not as evident.

## 4 RELATED WORK

**Selective state space models.** Mamba Dao & Gu (2024); Gu et al. (2020); Gu & Dao (2024); Hasani et al. (2021) is a structured state-space model (SSM) Goel et al. (2022); Gu et al. (2022); Gupta et al. (2022); Smith et al. (2023); Gu & Dao (2024) optimized for efficient long-sequence processing through a selective mechanism. Traditional SSMs use static, time-invariant parameters, which restrict their adaptability to changing sequence contexts. Mamba makes these parameters input-dependent through a selective mechanism, allowing adaptive filtering of relevant historical information based on the input sequence. This adaptivity helps Mamba prioritize key information dynamically, adjusting the timestep $\Delta(x_t)$ as a gating function to capture long-range dependencies better.

**Sequence modeling for RL.** Transformers are popular in RL due to their capacity to model long-term dependencies and capture complex patterns in sequential data via self-attention, which enables effective global context modeling. This capacity is essential for planning in large-scale environments. However, the quadratic computational and memory complexity of Transformers with respect to sequence length remains a drawback, especially for real-time applications that require processing extensive histories. In the realm of sequential decision-making, Decision Transformers reframe RL as a sequence modeling problem by treating states, actions, and rewards as a sequence. They use a Transformer to predict actions that maximize expected returns Chen et al. (2021). Variants of Decision Transformers enhance long-horizon planning by capturing global context from observations Fang et al. (2019). To improve efficiency, some works try to reduce the overhead through actor-learner distillation, which minimizes computational load during inference Parisotto et al. (2021). Transformer-based world models serve as visual planners, aiding agents in complex visual environments Tang et al. (2022). These models employ spatial attention to enable agents to focus on relevant spatial features for visual navigation tasks Wang et al. (2023). Some approaches also extend Decision Transformers to online settings, allowing models to adapt incrementally to new data Zheng et al. (2022).

Mamba-based architectures offer a compelling solution to the limitations of RNNs and Transformers, leveraging selective SSMs for efficient RL and planning. Research highlights Decision Mamba (DeMa) for its ability to handle long sequences with scalability advantages over Transformers, making it highly effective for sequence modeling Ota (2024). Studies on Mamba's compatibility with trajectory optimization confirm DeMa's strengths in offline planning Dai et al. (2024), while hybrid selective sequence modeling variants improve policy learning in complex environments Huang et al. (2024). Further extending its use, multi-grained state-space models with self-evolution regularization address the exploration-exploitation balance in offline RL Cao et al. (2024). In visual navigation, deep state-space models Krantz et al. (2023) enhance performance in memory-intensive tasks by effectively integrating past observations.

## 5 CONCLUSION AND FUTURE WORK

This paper presented a novel framework that enhances reward prediction fidelity in MBRL. By combining a multi-horizon reward ensemble, cross-horizon consistency regularization, and chunk-level reward modeling with slow-fast consistency, our approach addresses the critical challenge of reward prediction drift over extended horizons. Empirical results exhibit reduced reward prediction error and improved policy performance compared to the baseline Drama. It shows that explicitly modeling rewards at multiple temporal scales and enforcing consistency between them aid robust MBRL for more efficient policy learning, especially in environments requiring long-horizon reasoning. Future work could explore automatic, adaptive horizon selection for ensemble and more elegant consistency design instead of relying on manual, environment-specific configurations.

## REPRODUCIBILITY STATEMENT

An algorithm demonstrating the training workflow of the world model is shown in Algorithm 1. An architecture diagram that strictly reflects the true construction of the framework and the training and imagination procedures is exhibited in Fig. 2. The implementation of the core components of our proposed method can be found in Sec. A.7. More detailed implementation of training and inference can be found in Sec. A.8. We commit to open-source the full framework if the paper is accepted.

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

# A  APPENDIX

## A.1  THE USE OF LARGE LANGUAGE MODELS

Large language models (LLMs), including ChatGPT and Claude, are used to polish the writing, correct grammar errors, and refine the organization of some sections of this paper. Specifically, we use GPT-4o for writing refinement and grammar correction, and Claude Opus 4 for limited content organization for Sec. 2.

## A.2  HORIZON SET CHOICE AND PERFORMANCE

Different underlying reward feedback properties determine distinct horizon set selection strategies. A uniform set, e.g., $\{2, 4, 8, \ldots, 64\}$, does not always align with a task's reward pattern. We study the impact of task-aware horizon set selection in multiple representative Atari game environments. The results are displayed in Tab. 2 and Tab. 3. In *Gopher*, rewards arrive in rapid cycles of 3–5 frames, so denser short horizons help. Using a Fibonacci-like set $\{2, 3, 5, 8, 13\}$ or arithmetic sequence $\{2, 4, 6, 8\}$ lead to a higher score 3342 than 2520 corresponding to the uniform set. *DemonAttack* and *Krull* show a similar dependency on $\mathcal{H}$. This supports the view that awareness of the reward pattern is useful and that inaccurate reward estimates can degrade the policy quality.

Table 2: Effect of task-aware horizon sets. Scores are averages over independent seeds.

| Task | Baseline | Uniform $\mathcal{H}$ | Task-Aware $\mathcal{H}$ |
|---|---|---|---|
| DemonAttack | 201 | 198 | **361** |
| Gopher | 2757 | 2520 | **3342** |
| Krull | **9693** | 8712 | 9323 |

Table 3: Sensitivity to the choice of $\mathcal{H}$. Numbers are averaged over 5 seeds.

| Horizon Set $\mathcal{H}$ | Alien | Asterix | DemonAttack | Gopher | Krull |
|---|---|---|---|---|---|
| $\{2, 4, 8, \ldots, 64\}$ (uniform) | **912** | **1681** | 198 | 2520 | 8712 |
| $\{2, 3, 5, 8, 13\}$ | 885 | 1398 | 183 | **3342** | 8488 |
| $\{4, 8, 12, 20, 32\}$ | 868 | 1472 | **361** | 2586 | 8695 |
| $\{5, 10, 20, 40, 80\}$ | 830 | 1665 | 147 | 2374 | **9374** |

## A.3  EVALUATION PROTOCOL AND AGGREGATE METRICS

To obtain the numbers shown in Tab. 1, we follow the Drama's setup that uses 5 independent seeds per Atari game with means over 5 episodes. For a stronger aggregate view, we compute IQM, mean, median, and optimality gap on 7 representative games using 5 seeds per game and task-aware $\mathcal{H}$ where applicable. These preliminary results indicate consistent directional improvement, but with overlapping confidence intervals due to the small sample.

Table 4: Aggregate metrics on 7 games with 5 seeds each. Higher is better except optimality gap.

| Metric | Drama | Ours | Difference | Statistical Signal |
|---|---|---|---|---|
| IQM $\uparrow$ | 0.42 $[0.31, 0.58]$ | 0.48 $[0.35, 0.67]$ | $+14.3\%$ | overlapping |
| Mean $\uparrow$ | 1.33 $[0.89, 1.84]$ | 1.47 $[1.01, 2.02]$ | $+10.1\%$ | overlapping |
| Median $\uparrow$ | 0.30 $[0.17, 0.84]$ | 0.31 $[0.18, 0.98]$ | $+5.4\%$ | overlapping |
| Optimality Gap $\downarrow$ | 0.67 $[0.42, 0.83]$ | 0.53 $[0.33, 0.73]$ | $-20.9\%$ | overlapping |

These patterns motivate a full rerun on all 26 games with 5 seeds, performance-profile plots, and stratified bootstrap for game-level correlation.

Table 5: Efficiency comparison. Lower is better for times and memory; relative overhead reported vs Drama.

| Metric | Drama | STORM | Ours | Overhead vs Drama |
|---|---|---|---|---|
| World Model Update Time (ms) | 16 | 40 | 20 | +25% |
| Imagination Time (ms) | 2.4 | 9.0 | 3.6 | +50% |
| Total Inference per Step | baseline | 2.5×slower | 1.3×slower | +30% |
| Peak GPU Memory (GB) | 5.0 | 7.8 | 5.2 | +4% |
| Allocated GPU (GB) | 1.87 | 3.2 | 1.89 | +1% |
| Reserved GPU (GB) | 5.0 | 7.8 | 5.1 | +2% |

Table 6: Applying the method on a Transformer backbone (STORM).

| Method | Alien | KungFuMaster | MsPacman |
|---|---|---|---|
| STORM (baseline) | 984 | 26183 | 2673 |
| Ours on STORM | **1032** | **26710** | **2727** |

## A.4 EFFICIENCY EVALUATION

Our comprehensive benchmarking for computational requirements as shown in Tab. 5 reveals that while our approach does introduce overhead compared to the original Drama baseline, it remains significantly more efficient than Transformer-based alternatives. The numbers are based on the experiment in playing the game Alien for 5 runs with independent random seeds exactly the same as we used in the main paper. As the table shows, our method increases world model update time by only 25% (from 16ms to 20ms) and imagination time by 50% (from 2.4ms to 3.6ms) compared to Drama, while Transformer-based methods show 150% and 275% increases respectively. More evidently, the memory consumption remains nearly identical to Drama, with only a 4% increase in peak GPU memory (5.2GB vs 5.0GB), and negligible 1% and 2% increases for another 2 memory-related metrics. This demonstrates the memory efficiency of our multi-horizon ensemble despite using 7 reward heads. Furthermore, our curriculum learning strategy, which gradually increases the set of active horizons based on training progress, helps alleviate this cost during early training phases. Most importantly, this modest computational cost yields a 47% reduction in reward prediction error and 11% improvement in game scores, representing a favorable trade-off between computational efficiency and performance gains.

The facts about computational and memory costs also address potential concerns about building our method on the Mamba-based Drama rather than Transformer-based approaches such as STORM, which achieve higher policy performance. Our primary goal is to investigate the efficiency–performance trade-offs of using SSM backbones. While some Transformer methods earn higher scores, they do so at substantially higher computational and memory costs. Importantly, our method is not limited to SSMs. It can also be applied to Transformer-based models. As shown in Tab. 6, integrating our approach into STORM yields performance gains in several games.

## A.5 HYPERPARAMETER SENSITIVITY

We sweep loss weights in $[0.001, 1.0]$ on *Alien* with 5 seeds, varying one component at a time while fixing others to the baseline configuration (0.005 for multi-horizon, 0.001 for others). As shown in Tab. 7, performance is most sensitive to consistency terms. When their weights are large, optimization targets consistency rather than accuracy, which hurts scores. Multi-horizon and chunk losses are less sensitive but should remain down-weighted so as not to dominate the single-step reward objective.

## A.6 ROBUSTNESS TO VARIANCE MIS-CALIBRATION

We perturb predictive variances by factors of 0.1 (overconfidence) and 10 (underconfidence). Scores degrade by 6% and 3% respectively. Two factors explain robustness: diversity across 7 horizons

Table 7: Sensitivity of game score to loss weights on *Alien*. Larger is better.

| Weight | Multi-Horizon | Cross-Horizon Consistency | Chunk-Level | Slow-Fast Consistency |
|---|---|---|---|---|
| Reported | 912 | 912 | 912 | 912 |
| 0.000 | 624 | 821 | 873 | 885 |
| 0.001 | 890 | **912** | **912** | **912** |
| 0.005 | **912** | 795 | 860 | 776 |
| 0.010 | 840 | 579 | 694 | 588 |
| 0.050 | 781 | 323 | 626 | 509 |
| 0.100 | 663 | 230 | 563 | 313 |
| 0.500 | 512 | 135 | 455 | 150 |
| 1.000 | 345 | 101 | 241 | 90 |

reduces correlated failures, and the head-weighting function caps any single head's influence with a bounded weight controlled by $\varepsilon$ and $\beta$.

### A.7 NETWORK ARCHITECTURE

The architecture represents a Mamba-based model-based reinforcement learning framework with multi-horizon reward prediction and cross-horizon consistency. It consists of two main components: **(1) a world model** that learns to predict environment dynamics and rewards across multiple time horizons, and **(2) an agent module** that uses these predictions to learn optimal policies. We demonstrate the composition of our framework in details as follows.

#### A.7.1 WORLD MODEL

The world model encodes observations, predicts dynamics, and generates multi-horizon reward predictions through several specialized components.

**Perception components.**

- **Encoder.** A convolutional neural network (CNN) that transforms raw image observations ($3 \times 64 \times 64$) into latent feature representations.

- **Distribution head.** An output layer (head) that transforms features into categorical latent state distributions through a learned prior and posterior with uniform noise mixing.

**Dynamics model.** The sequence model using Mamba-2 as backbone, which processes sequences of latent states and actions to predict the next latent states. Specifically, it is a 2-layer Mamba with hidden dimension $512$ and state dimension $16$.

**Prediction heads.**

- **Image decoder.** Transposed CNN that reconstructs observations from latent states. It is trained by minimizing the reconstruction loss, which is the MSE between predicted and actual observations.

- **Single-step reward predictor.** MLP that outputs a SymLog Two-Hot encoded distribution for immediate rewards. It is trained by minimizing the SymLog Tow-hot loss between predicted and ground-truth immediate reward sequences.

- **Termination predictor.** MLP that outputs termination probabilities. It is trained by BCE loss between predicted and ground-truth termination sequences.

- **Multi-horizon cumulative reward predictors.** An ensemble of unique predictors for cumulative rewards over a set of candidate horizons. Each of these predictors is trained by minimizing SymLog Two-Hot loss between predicted cumulative reward and the real sum of multi-step reward for the corresponding horizon. We have tested multiple horizon set $\mathcal{H}$. See Sec. A.10 for details.

- **Chunk-level reward predictor.** MLP that outputs a SymLog Two-Hot encoded distribution for chunked rewards. It is trained by minimizing the SymLog Two-hot loss between the predicted reward sequence and real immediate reward sequence pooled every $C$ steps.

**Cross-horizon consistency mechanism.** We enforce that for any triplet $(h_i, h_j, h_j - h_i)$ where all are valid horizons belong to $\mathcal{H}$, Eq. (3) holds. We implement this regularization through consistency pairs $(h_i, h_j)$ via appropriate temporal alignment as follows.

```
x_horizon_consistency_loss = 0.0

for hi, hj in active_consistency_pairs:
    head_i   = self.multi_reward_heads[f"reward_head_hi"](dist_feat)
    head_j   = self.multi_reward_heads[f"reward_head_hj"](dist_feat)
    head_diff= self.multi_reward_heads[f"reward_head_hj -
    ↪  hi"](dist_feat)

    pred_i   = decode_logits(head_i)    # with shape (B, L)
    pred_j   = decode_logits(head_j)    # with shape (B, L)
    pred_diff = decode_logits(head_diff) # with shape (B, L)

    # only compare over t=0..L-hj
    Lj = pred_j.shape[1] - hj
    Pi = pred_i[:, :Lj]
    Pj = pred_j[:, :Lj]
    Pd = pred_diff[:, hi:hi+Lj]  # shift by hi
    c_loss = F.mse_loss(Pj, Pi + Pd)
    x_horizon_consistency_loss += c_loss * (self.lam ** hj)
```

**Slow-fast consistency mechanism.** We enforce that for any chunked reward prediction $\hat{r}_t^{\text{chunk}}$ and the $C$-step sum of ground-truth per-step returns $\sum_{k=0}^{C-1} r_{t+k}$, Eq. (6) holds. We implement this regularization via temporal alignment below.

```
slow_loss = torch.tensor(0., device=dist_feat.device)

for i, end_idx in enumerate(end_idxs.tolist()):
    # slow_pred: the slow-head's decoded chunk-sum at chunk i
    slow_pred = self.symlog_twohot_loss_func.decode(logits_slow[:,
    ↪  i])
    # pick out the fast-head's C-step window starting at end_idx
    C = self.slow_chunk_length
    start = end_idx - (C - 1)
    fast_pred = self.symlog_twohot_loss_func.decode(logits_fast[:,
    ↪  start])
    # penalize the squared-error between the two:
    c = F.mse_loss(slow_pred, fast_pred)
    slow_fast_consistency_loss += c
```

A.8 DETAILS IN TRAINING THE NETWORK

**Curriculum learning for horizon complexity.** We implement a progressive curriculum for introducing multi-horizon reward prediction heads during training. Rather than simultaneously training all horizon predictors, we gradually increase the set of active horizons based on training progress. The implementation details are as follows.

```
self.active_horizons = [min(self.horizons)]  # Start with shortest
↪  one
self.horizon_curriculum_steps = [
(10000, self.horizons[:2]),  # After 10k steps, use first 2 horizons
(30000, self.horizons[:3]),  # After 30k steps, use first 3 horizons
(50000, self.horizons),      # After 50k steps, use all horizons
]
```

This approach allows the model to first master shorter-horizon predictions, which are inherently more accurate, before tackling the complexity of longer-horizon predictions. By incrementally increasing prediction difficulty, the model develops a more stable foundation for learning.

**Adaptive loss weighting.** We employ dynamic weighting for multi-horizon and consistency losses through a gradual ramp-up schedule based on training progress. The implementation details are as follows.

```
810    def get_adaptive_loss_weights(self, global_step):
811        warmup_steps = 5000
812        max_steps = 50000      # Full ramp-up by 50k steps
813
814        # No multi-horizon during initial warmup
815        if global_step < warmup_steps:
           return 0.0, 0.0
816
817        # Calculate progress factor (0 to 1)
818        progress = min(1.0, (global_step - warmup_steps) / (max_steps -
819    ↪   warmup_steps))
820        # Sigmoid ramp-up for smoother transition
821        ramp_factor = 1.0 / (1.0 + np.exp(-10 * (progress - 0.5)))
822
823        # Scale weights
824        multi_horizon_weight = self.multi_horizon_reward_weight *
       ↪   ramp_factor
825
826        # Extra gradual ramp for consistency losses
827        consist_ramp = progress * progress  # Quadratic ramp
       consistency_weight =
828    ↪   self.x_horizon_consistency_weight * ramp_factor *
829    ↪   consist_ramp
830
831        return multi_horizon_weight, consistency_weight
```

Starting with zero weight and gradually increasing prevents these additional components from destabilizing early training. The sigmoid-based ramp provides a smooth transition, while the quadratic ramp for consistency ensures this more stringent constraint is applied even more gradually.

**Multi-horizon reward ensemble mechanism.** We implement an advanced uncertainty-aware ensemble mechanism that combines predictions from multiple horizon heads based on three key factors: prediction confidence, horizon length, and temporal relevance. This approach creates a more robust reward prediction signal for policy learning by dynamically weighting predictions based on their reliability. The implementation details are as follows.

```
841    def get_n_step_predictions(self, dist_feat_buffer, batch_length):
842        # Get predictions from all active horizon heads
       per_head_logits =
843            h: self.multi_reward_heads[f"reward_head_h"](
844            dist_feat_buffer[:, :batch_length]
845            )
846            for h in self.active_horizons
847        per_head_values =
848            h: self.symlog_twohot_loss_func.decode(logits)
           for h, logits in per_head_logits.items()
849
850
851        # Compute prediction uncertainty using entropy
852        head_uncertainties =
853            h: -torch.sum(
854            F.softmax(logits, dim=-1) *
855            torch.log(F.softmax(logits, dim=-1) + 1e-8), dim=-1)
           for h, logits in per_head_logits.items()
856
857        # Apply multiple weighting factors and combine predictions
858        # Factor 1: Confidence factor
859        uncertainty_factor = torch.exp(-uncertainty_tensor)
860        # Factor 2: Horizon length factor
861        horizon_lengths = torch.tensor([
862            1.0/h for h in self.active_horizons
863        ])
           # Factor 3: Temporal relevance factor
           temporal_factor = torch.tensor([
```

```
        self.gamma ** (h / 10.0)
        for h in self.active_horizons
    ])
```

This ensemble mechanism incorporates three key factors in determining prediction weights. Firstly, regarding prediction uncertainty, we use the entropy of the predicted distribution as a direct measure of uncertainty, giving more weight to confident predictions. Secondly, for the horizon length, shorter horizons naturally provide more accurate predictions of near-term rewards, so they receive higher base weights. Finally, we apply an exponential discount factor based on the horizon length, acknowledging that predictions further into the future should have a diminishing influence. Through this uncertainty-aware ensemble mechanism, our framework maintains a delicate balance between prediction diversity and reliability, creating a more robust reward signal for effective policy learning in complex environments.

$\lambda$-**return and N-step reward blending.** We combine traditional $\lambda$-returns with multi-horizon reward predictions using a weighted blend to create enhanced return estimates. The implementation details are as follows.

```
# Calculate lambda-return using standard method
lambda_return = calc_lambda_return(
    reward, value, termination, self.gamma, self.lambd
)

# Get multi-horizon n-step reward prediction from ensemble
n_step_reward = self.n_step_reward_hat_buffer

# Blend the two return estimates
blend = self.n_step_blend_alpha  # Typically set to 0.5
mixed_return = blend * n_step_reward + (1.0 - blend) * lambda_return

# Use mixed return for both value and policy optimization
value_loss = self.symlog_twohot_loss_func(raw_value[:, :-1],
←   mixed_return.detach())
norm_advantage = (mixed_return - value[:, :-1]) / norm_ratio
policy_loss = -(log_prob * norm_advantage.detach()).mean()
```

$\lambda$-returns have strong theoretical guarantees but can be myopic in complex environments. Multi-horizon predictions offer better long-term reward estimation but may be less accurate for immediate rewards. Blending combines the strengths of both approaches, creating a return estimate that balances accuracy and foresight.

## A.9    ADDITIONAL EVALUATION RESULTS

See Fig. 5.

## A.10    IMPORTANT HYPERPARAMETERS

See Tab. 8.

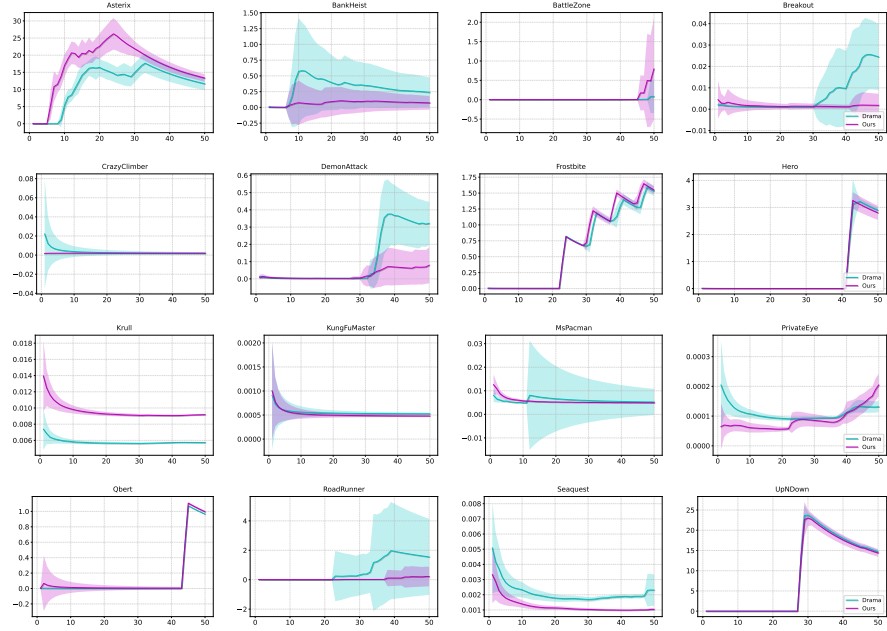

Figure 5: Visualized results of performance comparison between Drama and our approach in additional Atari environments. Average cumulative reward prediction errors against horizons.

Table 8: Important hyperparameters used for the framework.

| Component | Parameter | Value |
|---|---|---|
| Basic Settings | Image Size | $64 \times 64 \times 3$ |
| | Precision | BFloat16 (Mixed Precision) |
| World Model | Model Type | Mamba2 |
| | Hidden State Dimension | 512 |
| | Categorical Latent Dim | 32 |
| | Mamba Layers | 2 |
| | Mamba State Dimension | 16 |
| | Dropout | 0.1 |
| Multi-Horizon | Reward Horizons | task-aware |
| | Reward Weight | 0.005 |
| | Consistency Weight | 0.001 |
| | Curriculum Steps | $\{10000, 30000, 50000\}$ |
| Chunk-Level | Chunk Length | 8 or 10 |
| | Chunked Reward Weight | 0.001 |
| | Slow-Fast Consistency Weight | 0.001 |
| Behavior Policy | Paradigm | Actor-Critic |
| | Gamma | 0.985 |
| | Lambda | 0.95 |
| | Entropy Coefficient | $3 \times 10^{-4}$ |
| | Learning Rate | $4 \times 10^{-5}$ |
| | Actor Hidden Units | 256 |
| | Critic Hidden Units | 512 |
| Training | Batch Size | 16 |
| | Batch Length | 128 |
| | Imagine Batch Size | 1024 |
| | Imagine Context Length | 8 |
| | Imagine Batch Length | 32 |
| | Reality Context Length | 32 |
| Optimization | Optimizer | LaProp |
| | Learning Rate | $4 \times 10^{-5}$ |
| | Weight Decay | $1 \times 10^{-4}$ |

