# OpenReview forum: "Robust Reward Sequence Modeling with Multi-Scale Consistency for Model-Based Reinforcement Learning"
_ICLR.cc/2026/Conference — ICLR 2026 Conference Withdrawn Submission_

### Official Review · Reviewer_JEBH · 2025-10-26

**Soundness:** 3
**Presentation:** 2
**Contribution:** 2
**Rating:** 2
**Confidence:** 4

**Summary:**

This paper proposes a methodology to improve reward modeling in MBRL, building upon Mamba-based sequence models.
The approach consists of three main components: 1) Multi-step reward decoder ensemble, 2) Cross-horizon consistency, 3) Slow-fast consistency loss with a chunk-level predictor.
Empirically, the method reduces reward prediction error by an average of 47% and achieves higher scores in Atari100k in Mamba-based MBRL.

**Strengths:**

**S1. Clear improvement in performance**
* The proposed modifications leads to a noticeable reduction in reward prediction error and improved downstream performance (HNS mean 105 -> 116).
* The paper demonstrates that improved reward modeling can lead to stronger RL performance.

**Weaknesses:**

**W1. Contribution and novelty is not clear**

The work builds on several existing ideas; most notably, the multi-step reward decoder ensemble, which is similar to the technique introduced in CBOP [1]. It would help readers if the authors could:
* Explicitly discuss how their formulation differs from or extends CBOP (e.g., architectural, theoretical, or empirical distinctions).
* Emphasize which component(s) are newly introduced and most responsible for the observed improvements.
* If possible, include an ablation or discussion showing unique synergies among the three components.
Additionally, the slow-fast consistency loss appears to have relatively minor impact in the current results (Figure 4).
The authors could either strengthen its justification (perhaps by highlighting cases where it contributes meaningfully), or simplify the method if it is not essential.

**W2. Too much components and tuning.**
* While there is a substantial performance improvement, it adds 3 components and non-trivial tuning to achieve those.
* For example, horizon set selection (limiting horizons in a human-defined set for every task) is highly task-specific, which makes this methodology hard to apply in new domains.

**W3. Experiments limited to Mamba-based RL (Drama)**
The current experiments are limited to Mamba-based RL (Drama). Although this focus is reasonable given the model’s recent popularity, the underlying issues of reward prediction and temporal consistency are general. To strengthen the paper’s impact and demonstrate broader applicability, I recommend:
* Evaluating (even briefly) on other MBRL frameworks, such as RSSM (DreamerV3[2]) or TSSM (STORM[3]), to show generality across architectures.
* Alternatively, discussing how the proposed methods would generalize beyond Mamba-based models, possibly supported by a small-scale experiment or analysis.

[1] Jihwan Jeong et al., Conservative Bayesian Model-Based Value Expansion for Offline Policy Optimization, ICLR 2023.

[2] Danijar Hafner et al., Mastering Diverse Domains through World Models, Nature.

[3] Weipu Zhang et al., STORM: Efficient Stochastic Transformer based World Models for Reinforcement Learning, NeurIPS 2023.

**Questions:**

**Q1. Applicability to other model-based RL models**
* Since the proposed methods appear agnostic to the specific state-space model (SSM) used, have you considered testing them in RSSM or TSSM frameworks?
* This could help establish that the approach is not limited to Mamba-based architectures.

**Q2. Sensitivity to horizon set choice**
* Could you report results using a uniform horizon set across tasks?
* This would help clarify how much of the performance improvement depends on manual tuning and provide a fairer comparison with existing baselines.

---

### Official Review · Reviewer_pxJn · 2025-10-27

**Soundness:** 3
**Presentation:** 2
**Contribution:** 2
**Rating:** 4
**Confidence:** 4

**Summary:**

This paper proposes a Mamba-based MBRL framework which features a multi-scale reward modeling approach built on top of the Drama baselines. The authors developed an ensemble of multi-horizon reward heads, and a cross-horizon consistency loss that regularizes these heads by enforcing that their predictions are temporally consistent. Moreover, they further incorporate a "slow" chunk-level reward model that predicts total rewards for non-overlapping blocks. During policy training, an uncertainty-aware imagination mechanism is adopted that dynamically weights the predictions from the different reward heads during policy rollouts, relying more on heads with lower predictive variance. The method is evaluated on the Atari 100k benchmark, where it demonstrates a 47% reduction in reward prediction error and achieves a median normalized score double that of its baseline, Drama.

**Strengths:**

1. The paper tackles a well-known and critical problem in MBRL. The core idea of using a cross-horizon consistency loss to tie together an ensemble of multi-horizon predictors is novel, elegant, and provides an intuitive way to regularize the model's understanding of temporal dynamics. The uncertainty-aware ensemble for weighting predictions during imagination is also a very strong and sensible design choice.
2. The proposed method is a clear and logical extension of a strong, relevant baseline (Drama). The experimental evaluation is thorough and provides compelling evidence for the method's effectiveness.
3. The paper is well-written.

**Weaknesses:**

1. Appendix A.2 reveals that the set of horizons is not general but must be manually "task-aware" (i.e., hand-tuned) for each environment. The performance difference is not trivial; for example, DemonAttack scores 361 with a task-aware set but only 198 with a uniform set. This heavy reliance on per-game hyperparameter tuning significantly limits the method's generality and practical applicability as a general-purpose agent. This is a major limitation of the paper.
2. The ablation study is confusing with respect to the chunk-level coarse-grained reward modeling and slow-fast consistency components. Adding this part alone to Drama leads to the second highest error, while the earned score is the second lowest. This suggests the component is not very effective. The ablation study does not provide a clear picture of why this component is necessary or what value it adds in combination with the cross-horizon consistency module.
3. The claim of "higher or comparable game scores" seems to be made relative to the Drama baseline, not the entire field. In Table 1, the proposed method's mean score (116%) is lower than several other SOTA methods (HarmonyDream 136%, DreamerV3 125%, STORM 127%). While the improvement over Drama is a strong contribution in its own right, the framing could be more precise about its performance relative to the absolute SOTA.
4. Figure 2 is over-complicated and hard to understand.
5. The computational cost of the proposed approach is not thoroughly discussed in comparion to the Drama baseline.

**Questions:**

1. The paper is empirically very strong, but it lacks theoretical analysis. The proposed consistency losses and the uncertainty-aware ensemble are presented as intuitive heuristics. Is there any theoretical justification for why this specific form of regularization (enforcing consistency across an ensemble) leads to a more accurate value estimation or a tighter bound on policy performance error compared to a single, well-calibrated reward model? For instance, does this framework guarantee a reduction in the variance of the value estimates?
2. The authors conducted experimets on the Atari100K benchmark. I was wondering whether the proposed approach can generalize to other benchmarks.
3. See weaknesses.

---

### Official Review · Reviewer_owMc · 2025-10-31

**Soundness:** 2
**Presentation:** 2
**Contribution:** 3
**Rating:** 4
**Confidence:** 4

**Summary:**

The paper proposes enhancements to the reward head within the world model of model-based reinforcement learning. First, it replaces a single reward head with an ensemble of multi-horizon reward decoders that predict cumulative returns. Second, it regularises these decoders to enforce cross-horizon consistency. Finally, it introduces a chunk-level head—a low-frequency “slow” predictor—intended to reduce high-frequency noise and provide robust, coarse-grained guidance for long-term planning. On Atari-100k, the authors report a 47% average reduction in reward-prediction error and improved performance over the Drama baseline.

**Strengths:**

The research question is well-motivated for the model-based reinforcement learning (MBRL) community. Reliable reward modelling is crucial because compounding prediction noise can degrade policy quality. The empirical results align with the paper’s objective: a reported 47% mean reward-error reduction and faster convergence. Although the main benchmark uses a Mamba-2-based world model, the method appears architecture-agnostic; appendix experiments with a Transformer-based world model—while not a full benchmark—show similar trends. The ablation study clearly demonstrates the effectiveness of multi-horizon heads and cross-horizon regularisation, whereas the chunk head offers only marginal benefits.

**Weaknesses:**

I will consider increasing my rating if the following concerns are addressed during the rebuttal:
1. Figure 1 (imagination horizon): I assume the x-axis denotes the imagination horizon (steps). The cumulative error appears similar for horizons ≤15, with benefits becoming visible only at longer horizons. This raises the question of whether performance gains are larger on tasks that require long imagination trajectories. An additional long-rollout stress test would strengthen the claim. Actionable: add comparison experiments on MiniGrid-`DoorKey-16x16` and `MultiRoom-N6`-under a long rollout setting.

2. The ablations suggest the chunk-level head provides only marginal gains. Does it offer other advantages, such as stabilising training or reducing variance? Please quantify.

3. Figure 3, What is the justification for the four games shown?

4. Figure 4, I believe it is from `Alien` right? Can you provide integrated plot with another three games. Use normalised scores so high-scoring games do not dominate, and justify the game choices.

Minor issues:
1. Figure 1 is missing an x-axis label.
2. Term 'chunk' is not introduced when first mentioned.
3. In the introduction the paper says 25% improvement in median normalised score over Drama, but the table 1 shows 100%, please reconcile.
4. Notation $T$ used in Algorithm 1 is sequence length and under equation 2 "$T$ denotes the temperature-scaling parameter" but in equation (2) there isn't a capital $T$?

**Questions:**

1. Do you observe a more smooth score distribution in the games where the score is sparse like `Freeway` or `PrivateEye`?
2. The numerical number of the predicted score, is it still matching the original score or is it a more score engineering techniques?
3. The horizon sets, is it possible to do an auto selector in the future research.

---

### Note · Authors · 2025-11-12

I have read and agree with the venue's withdrawal policy on behalf of myself and my co-authors.